# Multicriteria Analysis in Apiculture: A Sustainable Tool for Rural Development in Communities and Conservation Areas of Northwest Peru

Alexander Cotrina-Sanchez [1,2,*], Ligia García [1,*], Christian Calle [1], Fatih Sari [3], Subhajit Bandopadhyay [4], Nilton B. Rojas-Briceño [5,6], Gerson Meza-Mori [1], Cristóbal Torres Guzmán [1], Erick Auquiñivín-Silva [1], Erick Arellanos [1] and Manuel Oliva [1]

1 Instituto de Investigación para el Desarrollo Sustentable de Ceja de Selva (INDES-CES), Universidad Nacional Toribio Rodríguez de Mendoza de Amazonas, 342 Higos Urco, Chachapoyas 01001, Peru; christian.calle.epg@untrm.edu.pe (C.C.); gmeza@indes-ces.edu.pe (G.M.-M.); cristobal.torres@untrm.edu.pe (C.T.G.); erick.auquinivin@untrm.edu.pe (E.A.-S.); erick.arellanos@untrm.edu.pe (E.A.); manuel.oliva@untrm.edu.pe (M.O.)

2 Department for Innovation in Biological, Agri-Food and Forest Systems, Università degli Studi della Tuscia, Via San Camillo de Lellis, 4, 01100 Viterbo, Italy

3 Faculty of Engineering and Natural Sciences, Geomatic Engineering Department, Konya Technical University, Konya 42250, Turkey; fsari@ktun.edu.tr

4 Geography and Environmental Science, University of Southampton, Southampton SO17 1BJ, UK; subhajit.iirs@gmail.com

5 Escuela Profesional de Ingeniería Ambiental, Facultad de Ingeniería y Arquitectura, Universidad Nacional de Moquegua, Pacocha 18610, Peru; nrojasb@unam.edu.pe

6 Escuela Profesional de Ingeniería Ambiental y Recursos Naturales, Facultad de Ingeniería, Universidad Tecnológica de los Andes, Abancay 03001, Peru

* Correspondence: alexander.cotrina@untrm.edu.pe (A.C.-S.); ligia.garcia@untrm.edu.pe (L.G.)

**Abstract:** Apiculture plays a vital role in maintaining a genetically diverse ecosystem and is an economic activity that contributes to the development of rural communities, thereby enhancing the livelihoods of beekeepers. However, despite the presence of over forty thousand beekeepers in Peru, there is currently no cartographic information available on optimal areas for the development of apiculture. Our study focused on assessing the suitability of land for apiculture development in rural and indigenous communities within the Amazonas Department in northwest Peru. We integrated biophysical and socioeconomic criteria using the Multiple Criteria Evaluation (MCE) technique, in conjunction with state-of-the-art geoinformation and earth observation techniques, to model and validate land suitability for supporting apiculture. It was identified that suitability is influenced by biophysical criteria (65%) and socioeconomic criteria (35%), resulting in highly suitable areas covering 315.6 km$^2$ within the territory of peasant communities, 128.4 km$^2$ within native communities, and an additional 41.4 km$^2$ within conserved areas. Furthermore, to validate our results, we combined the use of high-resolution satellite imagery and visits to artisanal producers. This research provides valuable insights for spatiotemporal land use planning, emphasizing apicultural activity as a driver of rural development and biodiversity conservation. Consequently, this study contributes as a management tool to promote apicultural activities as support for rural development and in local-level decision making.

**Keywords:** land; *Apis mellifera*; GIS; Apiaries; suitability

## 1. Introduction

Apiculture contributes to the development of rural communities by articulating local economies through the use, consumption, and sale of its derived products such as honey, propolis, pollen, wax, and royal jelly [1,2]. Furthermore, this activity plays a fundamental

role in crop pollination, contributing to food security and maintaining a friendly and genetically diverse ecosystem [3–6]. Additionally, during the COVID-19 lockdowns, apiculture was included as a new hobby, improving the mental health and quality of life of many individuals, which has become an invaluable social aspect [7]. In Peru, a National Apiculture Development Plan (NADP) was approved in 2015, with the purpose of planning, managing, and promoting the development of apiculture [8]. From this, it was identified that around forty thousand beekeepers work in approximately 300,000 beehives nationwide, allowing them to improve their economic and social condition through self-employment [9,10].

However, despite Peru's large number of geographical features, diverse climates, and forests, forest coverage has been reduced due to the intensification of agriculture and deforestation, which are the main threats faced by multiflora nature at the national level and in rural communities in northwestern Peru [9,11,12]. To recover these degraded areas, it is important to promote the regeneration of natural cover with native species of wild flora, provide food for pollinators and produce wild fruits to be used by rural communities [9,13]. In addition to the use of non-timber forest products, in rural areas, apiculture is a primary or secondary source of income, diversifying their income quickly in farms with little land and/or limited capital [14–17]. Nevertheless, in many cases, the installation of beehives is carried out empirically or traditionally without a comprehensive evaluation of the territory's potentialities and limitations. Therefore, proper land use planning from a spatiotemporal perspective will allow for the determination of suitable locations considering ecological, economic, social, and environmental aspects [18,19].

A Land Suitability Analysis (LSA) will contribute to building a solid foundation for the implementation of projects or activities such as apiculture, allowing for spatially appropriate decisions for beehive installation, increasing their yield and effectiveness based on physical, environmental, social, and economic data [2,3]. To identify potential sites, it is important to consider certain criteria and restrictions from topographical, environmental, meteorological, and economic perspectives, combining information collected from the field, expert opinions, and the use of technological tools such as Geographic Information Systems (GIS) and Remote Sensing (RS) [2,20,21]. However, freely accessible and low-resolution satellite images have limitations for the continuous monitoring of phenology and identifying coexisting plants where honey bees collect pollen and for delineating ecologically suitable areas for their habitat or beehive installation for breeding [2,18,22]. Therefore, a low-cost alternative for identifying suitable apiary locations is through the integration of GIS and Multiple Criteria Evaluation (MCE) techniques using the Analytic Hierarchy Process (AHP) [2,3,23–25]. This integration allows for the delineation of zones for honey bee breeding around the forest periphery, evaluation of the apicultural potential of pastures, and land suitability for the growth of meliponiculture [25–27].

The MCE technique is typically carried out for land suitability analysis or evaluation of a specific application, and through the AHP process, the suitability is determined based on weights assigned to evaluation criteria or sub-criteria [28–30]. The assigned weights represent the importance of the criteria and are compared with each other through a pairwise matrix before generating the suitability map [2,31]. MCE and AHP in integration with GIS and RS have been effectively used in site selection for apiculture, combining physical parameters (temperature, slope, flora, elevation, forage potential, distance to water resources, and power lines), economic parameters (distance to roads and market distance), and social parameters (land use and distance to urban areas) [2,3,25,27], GIS and Landsat and SPOT 5 images to identify potential locations for the production of propolis and honey [18,32]. Therefore, this integration complemented with field data through a proper validation process will contribute to the decision-making process for apicultural activities and can be replicated at a national or local scale [2,32].

Apiculture is a fundamental activity that contributes to improving the quality of life and health of people in these post-pandemic times [7] and is an alternative livelihood option with potential incentives for poor rural populations [33], as well as boosting the economy of small-scale producers in rural and indigenous communities like those in Amazonas, which

is one of the poorest departments in Peru [9,34]. In this respect, previous research in this area, applying Geographic Information Systems (GIS) and Multi-Criteria Decision Analysis (ACMC) to the biotic needs of bees and other important factors in apiary management, has allowed the generation of management tools in ecosystems in countries such as Malaysia (Selangor) [23], Turkey [35], India [24] and now in Peru. Therefore, the objectives of this study focused on (1) identifying suitable areas to promote the development of apiculture, using integrated biophysical and socioeconomic criteria through MCE and AHP techniques, and (2) validating the identification of highly suitable areas for apiculture through the combination of remote sensing techniques, GIS, and field visits to rural beekeepers in rural and indigenous communities in the northwest of the Amazonas department in Peru. Thus, this study contributes as a management tool to promote apicultural activities as support for the rural development and implementation of projects that contribute to improving the quality of life of local populations with a focus on protecting biodiversity.

## 2. Materials and Methods

### 2.1. Area of Study

The Amazonas department has an altitudinal gradient ranging from 120 to 4900 m above sea level (Figure 1), with temperatures ranging from 2° in the southern part to 40° in the northern part. It is in the northeast of Peru, between the parallels 2°59′12″ and 6°59′35″ south and the meridians 77°09′27″ and 78°42′06″ west, covering an approximate area of 40,000 km$^2$ [36]. In this territory, there are 59 rural communities and 362 native communities made up of Awajún and Wampis indigenous peoples [37]. Additionally, the Amazonas department is characterized by its high floristic and biological diversity distributed in its four types of forests (premontane, cloudy montane, inter-Andean, and lowland jungle) [38], highlighting the importance of an integrated evaluation of the communal territory and eco-physiographic factors to determine potential sites for the development of apiculture in this sector of the northwest of Peru.

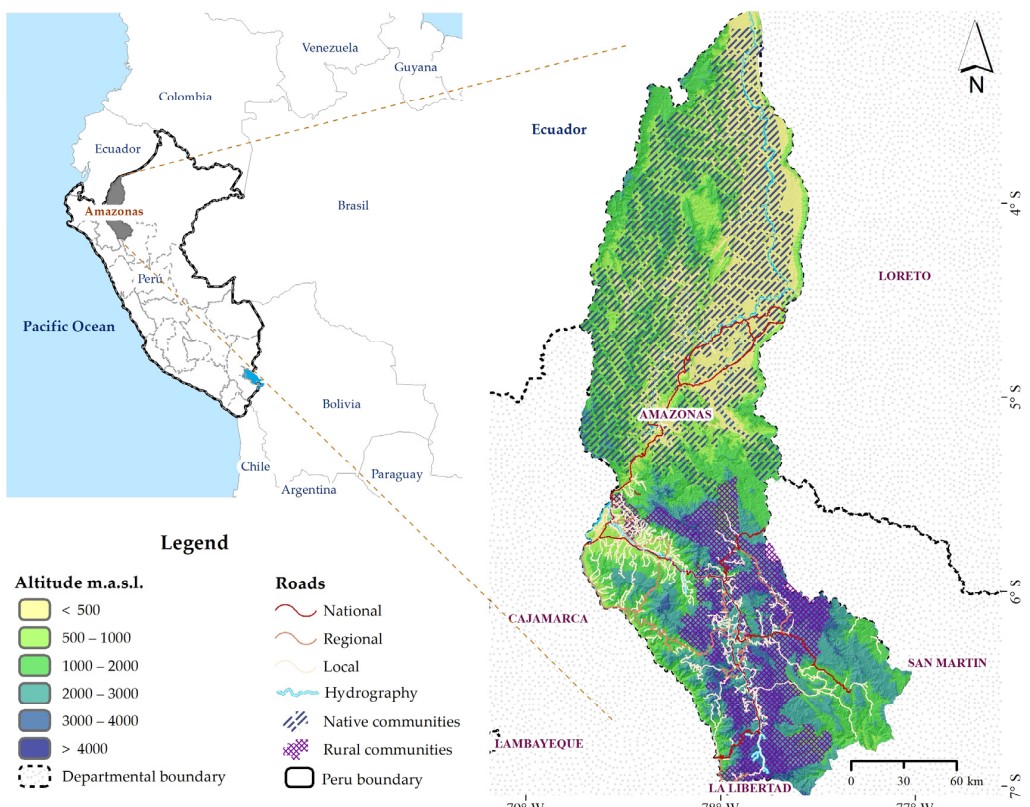

**Figure 1.** Location map of the Amazonas region.

### 2.2. Methodology

In this study, we adopted a synergetic approach to understand the suitability of Apiculture in the rural territories of the Peruvian Amazon.

Figure 2 shows the conceptual methodological diagram for identifying areas with beekeeping suitability in the department of Amazonas, integrating cartographic data with the use of GIS and RS, applying the Multiple Criteria Evaluation (MCE) and Analytic Hierarchy Process AHP, and subsequent validation of the previously reclassified areas according to their beekeeping suitability records.

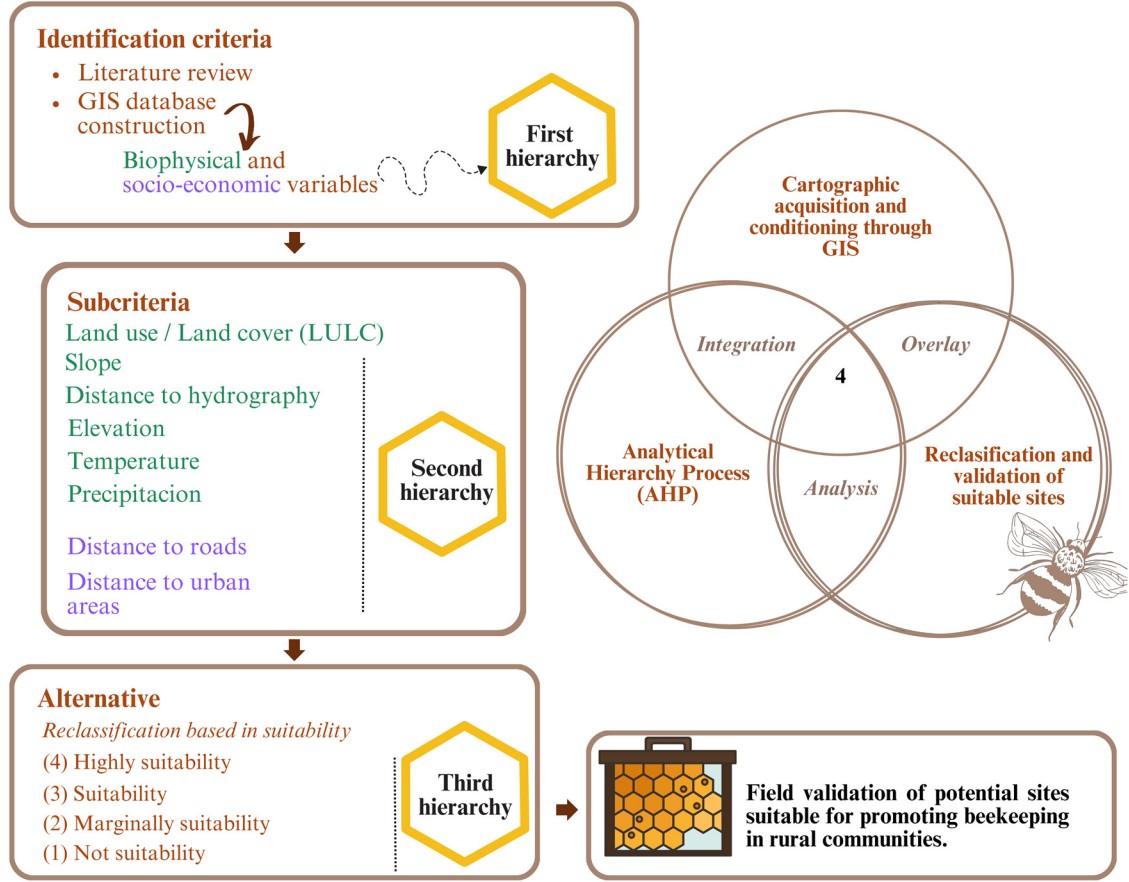

**Figure 2.** Conceptual methodological design.

### 2.2.1. Data Source

From the literature review and expert consultation, the factors influencing beekeeping activity were identified, considering biophysical and socioeconomic variables, conditions for hive management, and the biological requirements of bees [2,27,39]. Furthermore, for the validation of the highly optimal areas identified, high-resolution images (3 m) obtained from the PlanetScope satellite constellation were used. These satellites operate 24 h a day, 7 days a week, and collect weekly and even daily images worldwide, allowing for observation of land cover [40–42]. Table 1 describes the data sources used in this regional analysis, prioritizing freely accessible cartographic data, allowing for replication and/or adaptation according to the specific requirements and conditions of the area to be evaluated.

**Table 1.** Identification of suitable criteria and sub-criteria for beekeeping.

| Criteria/Variables | Description of Sub-Criteria | Source/URL |
|---|---|---|
| Biophysical | Land Use Land Cover (LULC) | https://dynamicworld.app/ (accessed on 15 March 2023). |
| | Digital Elevation Model (DEM) | [43] |
| | Slope (derived from DEM) | |
| | Temperature | [44] |
| | Precipitation | [44] |
| | Hydrography | https://snirh.ana.gob.pe/observatorioSNIRH/ (accessed on 10 March 2023). |
| Socioeconomic | Roads | https://portal.mtc.gob.pe/estadisticas/descarga.html (accessed on 10 March 2023). |
| | Urban areas | https://dynamicworld.app/ (accessed on 15 March 2023). |

To identify areas with adequate floral resources for beekeeping [45], current land use and land cover (LULC) maps with a resolution of 10 m were used, and areas with vegetation cover and crops that produce flowering for nectar and pollen production were filtered through cartographic analysis. Altitude, slope, and precipitation determine the distribution of forest species, floral composition, installation of crops, and consequently their relationship with honey production, although higher altitudes negatively contribute to apicultural suitability [39]. However, temperature is a fundamental factor in the phenology of flowering and insect development, playing a crucial role in their biology, so its consideration is essential in identifying optimal areas for poikilothermic organisms such as bees [46,47].

The Euclidean distance to water networks was obtained, where a higher value is associated with areas closer to streams, rivers or lakes; likewise, areas closer to roads are more suitable for the transport and installation of beehives [2,22,39]. As detailed information about the markets distributed in the region was not available, urban areas were considered [21,25] where retail sales of products derived from beekeeping are carried out.

Finally, the eight variables were standardized in raster format, with a spatial resolution of 30 m, and projected in a WGS 84 coordinate system, UTM Zone 18S. Figure 3 describes the subsequent processes of variable reclassification, weight assignment through AHP, obtaining sub-models, and finally, the final map of suitability for beekeeping. The latter was validated through high-resolution satellite imagery and field visits.

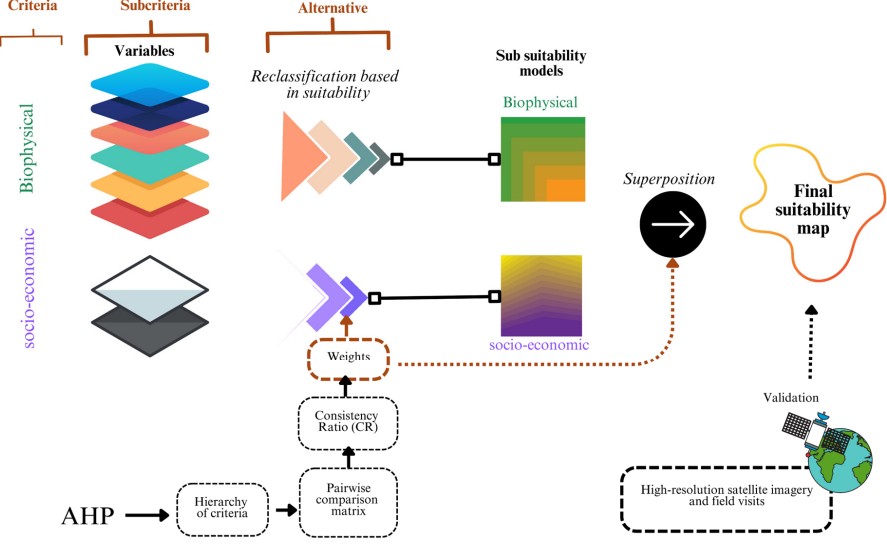

**Figure 3.** Methodological process.

### 2.2.2. Reclassification of Variables and Suitability Thresholds

To develop the third hierarchy (Figure 1), the sub-criteria were reclassified based on a literature review and expert opinions, allowing for the adaptation of variables according to the characteristics of the study area and the requirements of apiculture. Table 2 describes the suitability reclassification into 4 groups: Highly Suitable (4), Suitable (3), Marginally Suitable (2), and Not Suitable (1).

**Table 2.** Values of reclassification and hierarchical score of sub-criteria.

| Sub-Criteria/Variables | Highly Suitable (4) | Apt (3) | Marginally Adequate (2) | Not Suitable (1) | Adapted from the Studies: |
|---|---|---|---|---|---|
| **Biophysical variables** | | | | | |
| Land cover/Land use | Forest | Pastures and crops | Shrub/herbaceous vegetation | Bodies of water | [21,25,48] |
| Slope (%) | <5 | 5–20 | 20–45 | >45 | [35,49] |
| Distance to hydrography (km) | 0–0.5 | 0.5–1.5 | 1.5–3.0 | >3.0 | [2,27,35] |
| Elevation (m) | 200–1000 | 1000–1500 | 1500–2000 | 0–200/>2000 | [2,27,35] |
| Temperature (°C) | 20–25 | 15–20/25–27 | 10–15 | <10 | [25,49,50] |
| Precipitation (mm) | 1275–1800 | 1800–3000 | 3000–3500 | >3500 | [2,27,35] |
| **Socioeconomic variables** | | | | | |
| Distance to roads (km) | 0.5–1.5 | 1.5–2.0 | 2.0–3.0 | <0.5/>3.0 | [2,25,35,50] |
| Distance to urban areas (km) | >2.0 | 1.0–2.0 | 0.5–1.0 | <0.5 | [2,21,25] |

### 2.2.3. Analytic Hierarchy Process (AHP)

The AHP process is a decision-making tool that weights the priorities of each criterion and sub-criterion that affects the classification of the location of the apiary [2,25]. We conducted a weighting of the value judgments of 6 experts based on their experience and local reality, making a comparison between one criterion and another, constructing pairwise comparison matrices (PCMs), in which each variable compared to itself is equal to the unit diagonally (Table 3) [25,51]. For the AHP weighting process, we followed the proposal of Saaty (Table 4), with a scale from 1 (not important) to 9 (very important) [28,30].

**Table 3.** Pairwise comparison matrix.

| To | Criteria 1 | Criteria 2 | Criteria 3 | ... | Criteria nn |
|---|---|---|---|---|---|
| **Criteria 1** | 1 | $to_{12}$ | $to_{13}$ | ... | $to_{1n}$ |
| **Criteria 2** | $to_{21}$ | 1 | $to_{23}$ | ... | $to_{2n}$ |
| **Criteria 3** | $to_{31}$ | $to_{32}$ | 1 | ... | $to_{3n}$ |
| **...** | ... | ... | ... | 1 | |
| **Criteria nn** | $to_{n1}$ | $to_{n2}$ | $to_{n3}$ | | 1 |

**Table 4.** Importance value scale.

| Less important | | | | Equally important | More important * | | |
|---|---|---|---|---|---|---|---|
| Extreme | Strong | Moderate | | | Moderate | Strong | Extreme |
| 1/9 | 1/7 | 1/5 | 1/3 | 1 | 3 | 5 | 7 | 9 |

* 2, 4, 6, 8 or their reciprocally opposite values of lesser importance 1/2, 1/4, 1/6, and 1/8 are complementary intermediate values between two adjacent judgments.

The normalized matrix was generated by dividing each element by the sum of its column using Equation (1), and the average sum represented the weights using Equation (2) [2,28].

$$a_{ij}^1 = \frac{a_{ij}}{\sum_{i=1}^{n} a_{1j}} \tag{1}$$

$$w_i = \left(\frac{1}{n}\right) \sum_{i=1}^{n} a'_{ij'} \ (\text{i, j} = 1, 2, 3, \dots n) \tag{2}$$

However, the subjective preferences of experts can generate inconsistencies in prioritization during the construction of the PCM [52,53]. Therefore, the AHP method includes a consistency index (CI) to determine the reliability of the preference values given and decide whether the weighting calculations are consistent or not (Equation (3)) [28].

$$IC = \frac{\lambda max - n}{n - 1} \tag{3}$$

where "*n*" is the dimension of the matrix, and the other symbol is the maximum eigenvalue (λmax).

To calculate the *IC*, the *λmax* (eigenvalue) value is required, and it is necessary to compare the value of the *IC* with a random consistency index (*RI*). The *RI* is the average random index obtained from 500 randomly filled matrices previously calculated by Saaty [28,52], as shown in Table 5. Knowing the values of *IC* and *RI*, a consistency ratio (*CR*) is obtained (Equation (4)), and the value of *CR* indicates a reasonable level of consistency if it is within a *CR* < 0.1. Conversely, if *RC* > 0.1, it is an indicator that the judgments are inconsistent [23,30,54].

$$CR = IC/RI \tag{4}$$

**Table 5.** Random consistency index (RI) from [28,52].

| n | 1 | 2 | 3 | 4 | 5 | 6 | 7 | 8 | 9 | 10 |
|---|---|---|---|---|---|---|---|---|---|----|
| D | 0 | 0 | 0.58 | 0.9 | 1.12 | 1.24 | 1.32 | 1.41 | 1.45 | 1.49 |

2.2.4. Generation of the Suitability Model for Apiculture

The suitability model for beekeeping areas was generated using a linear weighted overlay in open-source spatial software through a summation of raster layers of the maps obtained because of the biophysical and socioeconomic sub-models, adding the criteria according to Equation (5).

$$grid_{result} = \sum_{i=1}^{n} (grid_1 * weight_i) \tag{5}$$

2.2.5. Validation of Results

Based on the obtained model of suitability for apiculture, a stratified random sampling of 10 different suitable areas was conducted for field visits. In areas with difficult access due to the rugged terrain, visual interpretation using high-resolution (3 m) multispectral satellite images obtained through PlanetScope (https://www.planet.com/products/monitoring/, accessed on 16 May 2023) was utilized. This approach allowed for a comparison of the identified suitable areas with existing apiculture installations and their locations classified through GIS and remote sensing.

## 3. Results

### 3.1. Criteria and Sub-Model Outputs

Based on the cartographic information collected and using the thresholds described in Table 2, the reclassified biophysical sub-criteria (Figure 4a–f) contributed to the biophysical suitability sub-model (Figure 4g). Similarly, from the reclassified socioeconomic sub-criteria (Figure 4h,i), the socioeconomic suitability sub-model was derived (Figure 4j).

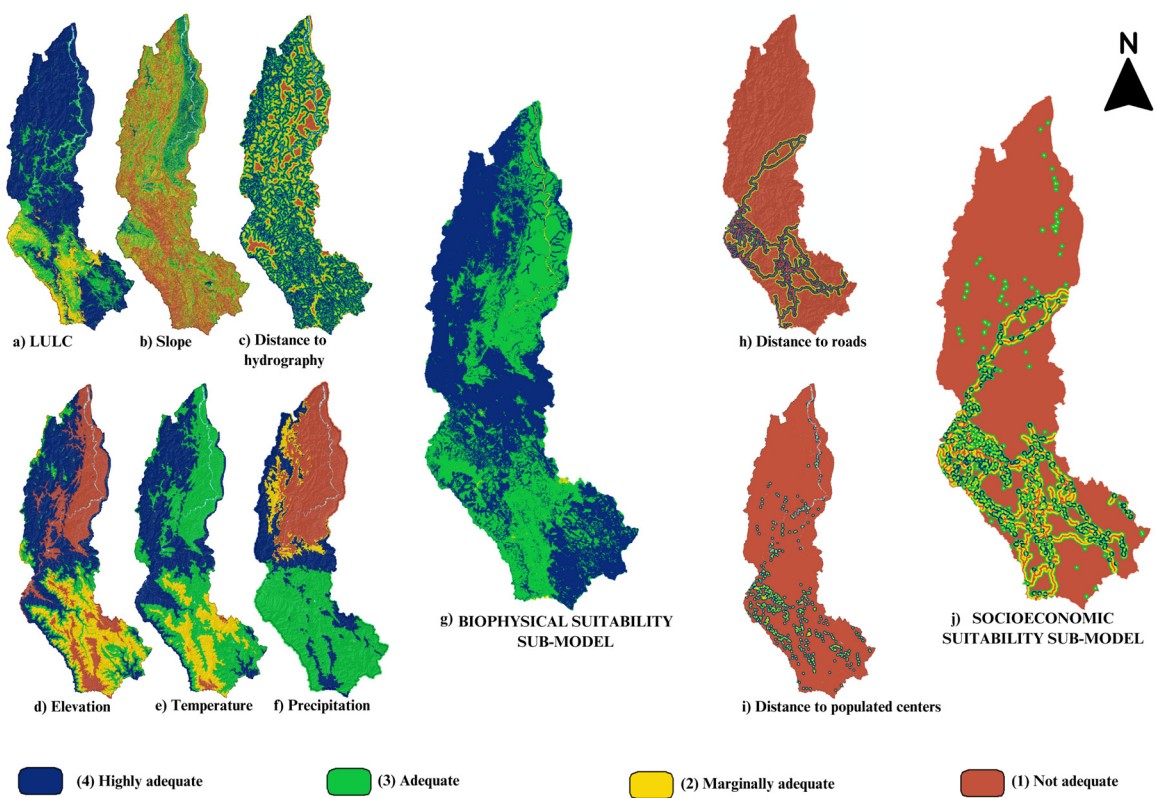

**Figure 4.** Reclassification of sub-criteria and obtaining sub-models.

Table 6 and Figure 5 describe and plot the surface percentages of each respective sub-criterion.

**Table 6.** Distribution areas of reclassified sub-criteria.

| Criteria | Sub-Criteria | Highly Suitable (km²) | Suitable (km²) | Marginally Adequate (km²) | Not Suitable (km²) |
|---|---|---|---|---|---|
| **Biophysical** | Land cover/land use | 28,306.2 | 8844.9 | 4356.3 | 75.8 |
| | Slope (%) | 2422.1 | 11,834.9 | 11,563.2 | 15,763.0 |
| | Distance to water systems (km) | 15,923.6 | 16,249.1 | 7045.4 | 2365.1 |
| | Elevation (m) | 15,598.4 | 5394.4 | 7917.1 | 12,673.2 |
| | Temperature (°C) | 16,041.4 | 19,769.6 | 5450.1 | 322.0 |
| | Precipitation (mm) | 8938.5 | 15,987.3 | 2940.4 | 13,717.1 |
| **Socioeconomic** | Distance to roads (km) | 2278.64 | 1448.18 | 4146.82 | 34,291.50 |
| | Distance to urban areas (km) | 3521.85 | 1413.68 | 910.31 | 36,221.76 |

Approximately 68% (28,306.2 km²) of the Amazonas department is covered by forest vegetation as part of the LULC, providing highly suitable conditions for beekeeping activity, in addition to presenting highly suitable temperatures ranging between 20 and 25° C, covering 38% of the study area. Additionally, a wide distribution of water networks with distances less than 1.5 km is considered suitable, and highly suitable areas cover 77% of the territory. However, it is worth noting that 40% of the department (15,763 km²) has a very steep slope (>45°), which would limit the development of this activity (Figure 4b). Furthermore, the main concentration of population centers is in interconnected spaces with roads (Figure 4h,i). Therefore, due to the limited interconnectivity of land routes in the Amazonas department, more than 80% of the territory would not have highly suitable conditions for beekeeping activity within socioeconomic considerations.

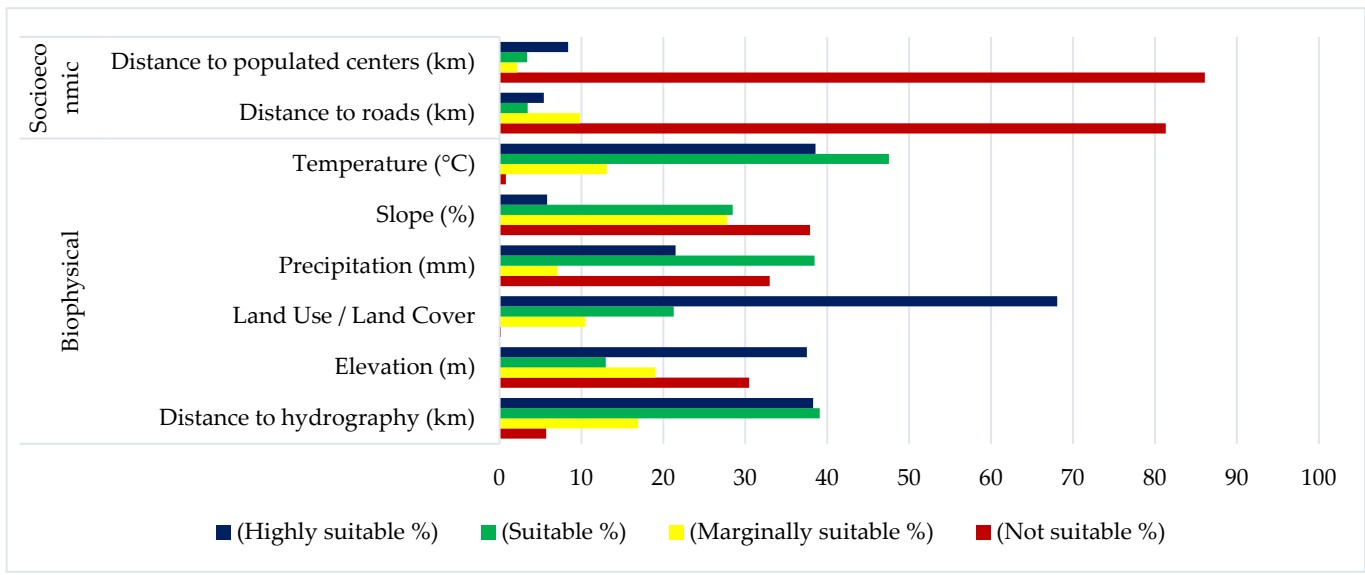

**Figure 5.** Percentage of coverage of criteria and sub-criteria.

### 3.2. AHP Outcomes

Table 7 shows the weightings of importance for the criteria and sub-criteria of suitability. Based on the AHP weightings obtained, 37% of the experts agreed that LULC determines the suitability of the land for beekeeping, which is followed by distance to water systems (20%) and precipitation (15%) for the biophysical criteria. Similarly, when weighting the socioeconomic criteria, distance to urban areas was found to be the most important sub-criterion (56%) compared to distance to roads (44%).

**Table 7.** Weightings of importance for the criteria and sub-criteria of suitability.

| Objective | Criteria | Sub-Criteria (Figure) | Weight |
|---|---|---|---|
| Areas suitable for beekeeping | Biophysical | LULC (Figure 4a) | 0.37 |
| | | Slope (Figure 4b) | 0.06 |
| | | Distance to hydrography (Figure 4c) | 0.20 |
| | | Elevation (Figure 4d) | 0.12 |
| | | Temperature (Figure 4e) | 0.10 |
| | | Precipitation (Figure 4f) | 0.15 |
| | Socioeconomic | Distance to roads (Figure 4h) | 0.44 |
| | | Distance to populated centers (Figure 4i) | 0.56 |

In addition, an RC value of 0.07 was obtained (Table 8), which is an acceptable and reliable coefficient, based on the averages of the pairwise comparison matrices (Supplementary Table S1).

**Table 8.** Average consistency ratios of criteria and sub-criteria.

| Proportion | Biophysical Sub-Criteria | Socioeconomic Sub-Criteria | Eligibility Criteria |
|---|---|---|---|
| N | 6 | 2 | 2 |
| Λmax | 6.45 | - | - |
| IC | 0.09 | - | - |
| AI | 1.25 | 0.0 | 0.0 |
| RC | 0.07 | | |

### 3.3. Sub-Models and Final Suitability Model

The biophysical (Figure 4g) and socioeconomic (Figure 4j) sub-models were obtained considering the weights described in Table 7 through the following equations:

$$
\begin{aligned}
Biophysical\ submodel_{grid} \\
= Grid_{LULC} * 0.37 + Grid_{slope} * 0.06 + Grid_{Dist.\ hydro} * 0.2 + Grid_{elev.} * 0.12 + Grid_{T°} * 0.1 \\
+ Grid_{precip.} * 0.15
\end{aligned}
$$

$$
Socioeconomic\ submodel_{grid} = Grid_{Dist.\ roads} * 0.44 + Grid_{Dist.\ urban} * 0.56 \tag{6}
$$

The highly suitable biophysical conditions cover 56.5% (24,168.1 km$^2$) of the territory, while the socioeconomic sub-model covers 4.6% (1943.2 km$^2$) and is categorized as having very high suitability for apiculture. Finally, the overlay of sub-models allowed the identification of optimal areas for apiculture in the department of Amazonas (Figure 6e), which is based on the following:

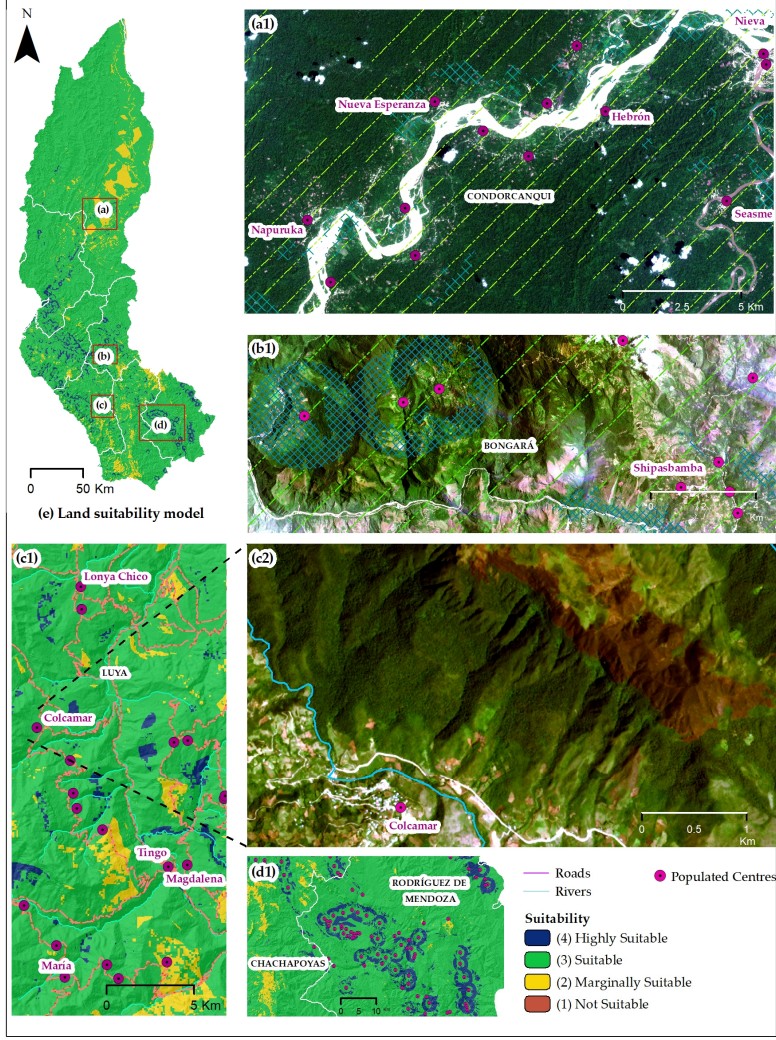

**Figure 6.** Suitable areas for the development of beekeeping in the provinces of Condorcanqui (**a**), Bongará (**b**), Luya (**c**), and Rodriguez de Mendoza (**d**). In territories of native communities (**a1**), rural communities (**b1,c1,c2**), and private properties (**d1**).

It was identified that there is approximately 1607 km$^2$ with highly suitable conditions for beekeeping development in the department of Amazonas, and the largest territory is considered suitable (89.5%). Table 9 describes in detail the suitability surfaces at the

provincial level, with the Utcubamba and Rodríguez de Mendoza provinces having the highest highly suitable surfaces.

**Table 9.** The area at km² and % for beekeeping.

| Provinces | Highly Suitable | Adequate | Marginally Adequate | Total |
|---|---|---|---|---|
| Bagua | 299.1 | 5367.0 | 126.0 | 5792.2 |
| Bongará | 180.2 | 2726.1 | 112.9 | 3019.2 |
| Chachapoyas | 116.3 | 3763.8 | 621.7 | 4501.8 |
| Condorcanqui | 46.4 | 16,307.7 | 1463.2 | 17,817.3 |
| Luya | 200.5 | 2661.9 | 236.9 | 3099.3 |
| Rodriguez de Mendoza | 370.6 | 3255.1 | 86.6 | 3712.3 |
| Utcubamba | 393.6 | 3442.0 | 136.4 | 3972.1 |
| Overall total | 1606.8 | 37,523.6 | 2783.8 | 41,914.1 |
| % of coverage of study area | 3.8 | 89.5 | 6.6 | 100.0 |

In addition, high-resolution images allowed for validation that the distribution of highly suitable areas is located near populated centers within native communities (Figure 6(a1)), rural communities (Figure 6(b1)), and private properties (Figure 6(d1)). In addition, the impact on vegetation cover caused by forest fires was identified (Figure 6(c2)).

*3.4. Apiculture Suitability in Communal Areas and Conservation*

Figure 7 and Table 10 describe the areas of beekeeping suitability in the rural communities, native communities, and conserved areas. A total of 315.6 km² present highly suitable conditions in rural communities and 128.4 km² in native communities, respectively (Figure 7a). Similarly, private conservation areas (27.4 km²) and regional conservation areas (13.5 km²) are the ones with the largest area with high aptitude for the development and implementation of beekeeping activities (Figure 7b).

**Table 10.** Area at km² and % fitness for beekeeping.

| | Communal Territory | | | | | | | |
|---|---|---|---|---|---|---|---|---|
| | Rural Communities | | | | Native Communities | | | |
| Suitability | High | Middle | Low | Total | High | Middle | Low | Total |
| Total (km²) | 315.6 | 5737.4 | 729.2 | 6782.2 | 128.4 | 15,891.1 | 1193.6 | 17,213.9 |
| % coverage of conserved area | 4.7 | 84.6 | 10.8 | 100.0 | 0.8 | 92.3 | 6.9 | 100.0 |
| | Conservation Categories | | | | | | | |
| Private Conservation Area | | | | | 27.4 | 1362.7 | 176.9 | 1567.1 |
| Regional Conservation Area | | | | | 13.5 | 584.9 | 29.8 | 628.2 |
| Natural Protected Area | | | | | 0.4 | 3732.6 | 125.0 | 3858.0 |
| Overall total | | | | | 41.4 | 5680.2 | 331.8 | 6053.3 |
| % | | | | | 0.7 | 93.8 | 5.5 | 100.0 |

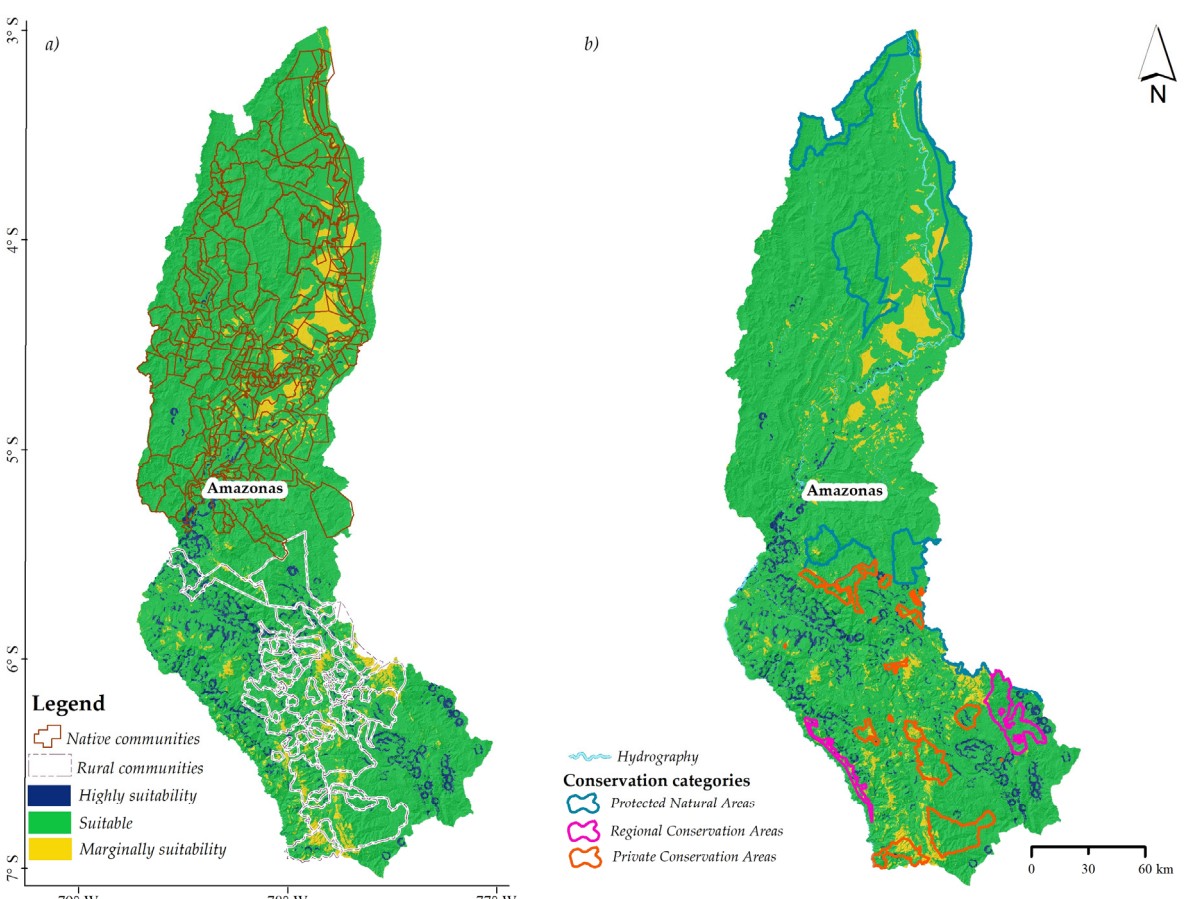

**Figure 7.** Suitability areas for beekeeping in community native and rural territories (**a**) and conserved areas (**b**) in the department of Amazonas.

Finally, through field visits, the artisanal beekeeping activities being implemented in the territory of the native communities sectors of Ideal and Pampa Hermosa in the Condorcanqui province (Figure 8(a1,a2)) were confirmed. Similarly, in the territory of the rural community of Shipasbamba (Figure 8(b1,b2)), Pampa Hermosa sector of Jumbilla district (Figure 8(c1–c4)), Taulia Molinopampa (Figure 8(e1,e2)), and the province of Chachapoyas (Figure 8d) were identified.

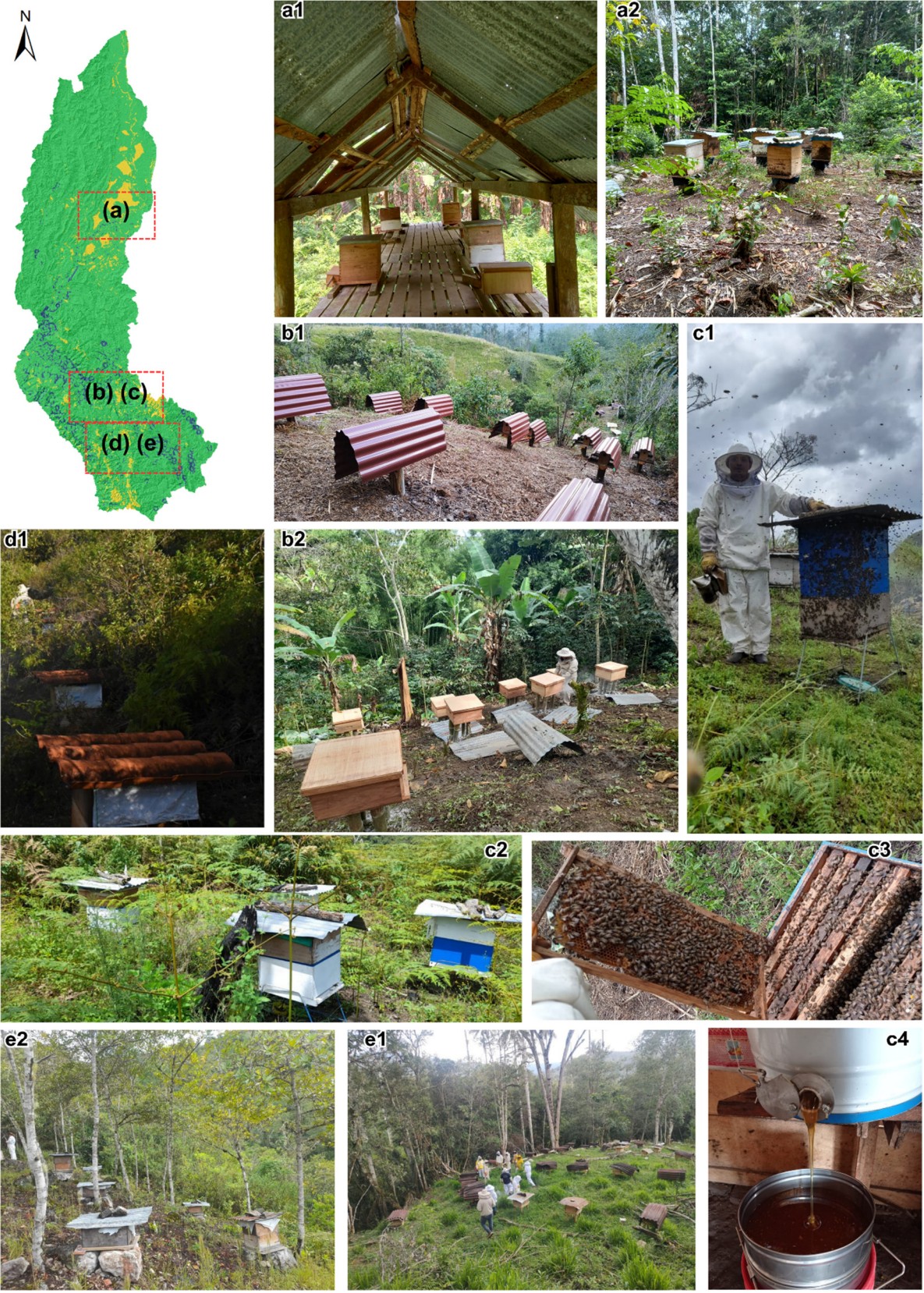

**Figure 8.** Development of artisanal beekeeping activities in the territory of native (**a1**,**a2**) and rural (**b1–e2**) communities. (**a**) native communities sectors of Ideal and Pampa Hermosa in the Condorcanqui province. (**b**) Shipasbamba. (**c**) Pampa Hermosa sector of Jumbilla district. (**d**) Opelel sector in Chachapoyas. (**e**) Taulía Molinopampa peasant community.

## 4. Discussion

Our study represents the first approach in identifying areas suitable for promoting rural beekeeping activity in the northwest of Peru, providing a tool for the implementation of current plans and projects for local development in coordination between the public and private sectors [8,10]. We used cartographic information by integrating GIS and remote sensing, grouping variables into biophysical and socioeconomic criteria. Of the latter, over 80% of the territory is located at a distance greater than 3 km from urban centers and roads, presenting low suitability for installing beehives, considering accessibility for the installation of bee colonies, honey transportation, and continuous monitoring [22,24,25,39]. The forest cover category of the LULC as part of the biophysical criterion constitutes 70% of the study area's coverage. However, between 2001 and 2021, 109,955.00 ha were lost [55] mainly for the installation of crops and pastures for extensive livestock that currently cover 21.7% (8844.9 km$^2$). Therefore, it is essential to consider that farmers are often unwilling to place beehives near their fields, even though three-quarters of crops benefit from insect pollination; thus, forested, shrubby, and grassy areas with a high diversity of plant species are preferred for beekeeping activity [35,56,57].

Using GIS and the integration with the MCE technique through the AHP process, we selected eight cartographic variables (sub-criteria) described in Table 7, considering that one of the main stages in the integration of these techniques in suitability analysis is the selection of criteria strictly related to the requirements needed for beekeeping activity [48]. However, the AHP weighting is subject to uncertainty caused by the influence of expert decision-makers [39]. Therefore, the consistency ratio (CR) is an indicator of judgment coherence, and a value of CR = 0.07 was obtained, which is a coefficient of a reasonable level [23,30,54]. Thus, the weighting values are within a range of reliability and are acceptable. It was identified that among the socioeconomic sub-criteria, the distance to urban areas predominates as a sub-criterion (56%) compared to the distance to roads (44%); the biophysical sub-criteria, which was mainly constituted by forests, pastures, and crops of the LULC (37%), would contribute in greater proportion to determining optimal areas followed by the distance to hydrography (20%) and precipitation (15%), respectively. This is consistent with previous studies where the importance of vegetation cover, vegetation composition, nectar and pollen-producing plants as fundamental factors for beekeeping is highlighted [25,48,50] in addition to temperature, elevation, and distance to markets, which are equally effective factors in determining a suitable suitability model [35,58,59].

One limitation of our study is the lack of available information on bee flora inventories in the Amazonas department [60], which could provide the foraging flight distance that depends on plant and flower density. This information has been considered in previous studies in Europe and southeast Asia [3,27,61]. However, in our study area, which is in the Andean and tropical region, it is expected that the foraging distances for bees to collect nectar and pollen will be shorter, considering that landscapes with diverse floral resources provide enough food for bees to avoid long travels [62] and that bees primarily seek food sources within a range of 1 km from their colony and up to 5 km for exceptionally rewarding sources [63,64].

Remote sensing (RS) contributes to validating and complementing studies related to beekeeping cartography through satellite images and vegetation indices [21,24,65]. For the validation of our results, we used high-resolution satellite images and field visits to the areas identified as having "highly suitable potential". This allowed us to verify in situ the reality of beekeeping carried out by the population settled in the rural and indigenous communities (Figure 8(a1–e2)), considering that beekeeping in Peru is not easy due to a lack of suitable technology, environmental awareness, and responsibility, which make this task a real challenge [66].

Therefore, our study constitutes a tool for decision making in the implementation of beekeeping activity, since many beekeeping intervention projects undertaken by local and regional governments as well as non-governmental organizations (NGOs) in Peru do not consider bee colony overpopulation in a particular area or the support of flora

in such spaces [67]. Finally, beekeeping is a sustainable opportunity for women's empowerment, such as that implemented by the Women's Committee of the Frontera San Ignacio Agricultural Cooperative (COOPAFSI), which is an association that stands out for its work in promoting beekeeping in the areas of the Andean bear corridor (Tabaconas Namballe National Sanctuary), San Ignacio-Cajamarca [68]. This is an exemplary activity to be considered for replication in agricultural cooperatives, conservation areas, and local communities in the Amazonas department.

## 5. Conclusions

In this research, we successfully combined efforts to identify optimal areas for promoting beekeeping activity in rural and indigenous communities in the northwest department of Amazonas, Peru. By employing the Multiple Criteria Evaluation (MCE) technique through the Analytic Hierarchy Process (AHP), we determined that these optimal areas are predominantly influenced (65%) by the biophysical sub-criteria (LULC, DEM, slope, temperature, precipitation, and aspect) followed by (35%) the socioeconomic sub-criteria (distances to roads and population centers).

These findings revealed a substantial land area of 1606.8 km$^2$ with highly suitable aptitude for apiculture in the Amazonas department, of which 315.6 km$^2$ pertain to rural communities and 128.4 km$^2$ pertains to native communities, respectively. Furthermore, potential for beekeeping activity was identified within private conservation areas (27.4 km$^2$) and regional conservation areas (13.5 km$^2$).

Finally, based on our results, high-resolution images and field visits in rural and communal territories were utilized to validate the obtained results and gain insights into the activities carried out by small-scale beekeeping producers. As such, this work, employing geospatial data, constitutes a pioneering study as a management tool for the implementation of projects at the local or regional level as well as a methodological framework that can be replicated and complemented in sectors related to decision making for proper territorial planning.

**Supplementary Materials:** The following supporting information can be downloaded at https://www.mdpi.com/article/10.3390/land12101900/s1, Table S1: Pairwise comparison matrix to evaluate the importance of sub-criteria and sub-models for the identification of suitability areas for beekeeping.

**Author Contributions:** Conceptualization, A.C.-S., C.C. and G.M.-M.; methodology, A.C.-S., C.C., G.M.-M., E.A., N.B.R.-B. and F.S. software, A.C.-S. and G.M.-M. validation, A.C.-S. and F.S.; formal analysis, A.C.-S., G.M.-M., L.G. and E.A. investigation, A.C.-S., G.M.-M., N.B.R.-B. and M.O.; resources, C.T.G., M.O., F.S. and S.B.; data curation, A.C.-S., C.T.G. and C.C.: writing—original draft preparation, A.C.-S., C.T.G., M.O. and F.S.; writing—review and editing, A.C.-S., E.A., F.S. and S.B.; visualization, C.T.G. and E.A. supervision, C.T.G., E.A.-S. and S.B.; project administration, C.T.G., L.G. and E.A.-S. funding acquisition C.T.G., M.O. and L.G. All authors have read and agreed to the published version of the manuscript.

**Funding:** The study was funded by the project CUI N° 2261386 "Creation of the Services of a Biodiversity Laboratory and Conservation of Genetic Resources of Wild Species of the Toribio Rodríguez de Mendoza National University, Amazonas"—BIODIVERSIDAD, and through the Instituto de Investigación para el Desarrollo Sustentable de Ceja de Selva (INDES-CES) in the project named "Creación de los Servicios del Centro de Investigación, innovación y Transferencia Tecnológica de Café de la Universidad Nacional Toribio Rodríguez de Mendoza de Amazonas" (C.U.I. N° 2317883—CEINCAFÉ).

**Data Availability Statement:** Not applicable.

**Acknowledgments:** The authors express their gratitude and acknowledge the support of the "Universidad Nacional Toribio Rodríguez de Mendoza de Amazonas" through the Institute for Sustainable Development of the Eyebrow of the Jungle (INDES-CES). Additionally, special thanks to external collaborators who provided access during visits and photography of their apicultural facilities.

**Conflicts of Interest:** The authors declare no conflict of interest.

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
