# Peer review of "Multicriteria Analysis in Apiculture: A Sustainable Tool for Rural Development in Communities and Conservation Areas of Northwest Peru"

_land, doi:10.3390/land12101900_

Round 1

Reviewer 1 Report

Apiculture is very important for rural development in Peru, but at other regions during the world. Apiculture can be interesting for the economy, but for touristic offers too why this article has a special value.

Abstract is very correct and key words are enough and good.

The first part Introduction is some long. Authors after introductions need to add part literature review, to take a closer look at previous research in this area.

All Figure sources are missing. Is it the work of the author or?

Methodology and results are presented very good with good tables and figures like as attachment to the explanation.

Conclusion too short and it needs to be expanded.

References are very numerous and contemporary, which is very good.

Author Response

Dear Reviewer,

We appreciate your suggestions and insightful comments, as well as the valuable time you have devoted to this work. We have now implemented the suggested modifications, all of which have been included in our revised manuscript submission.

Reviewer #1: Apiculture is very important for rural development in Peru, but at other regions during the world. Apiculture can be interesting for the economy, but for touristic offers too why this article has a special value.

Abstract is very correct and key words are enough and good.

The first part Introduction is some long. Authors after introductions need to add part literature review, to take a closer look at previous research in this area.

The introduction has been revised and the required previous studies to be considered have been included.

All Figure sources are missing. Is it the work of the author or?

All the figures are original and created by the authors.

Methodology and results are presented very good with good tables and figures like as attachment to the explanation.

Conclusion too short and it needs to be expanded.

The conclusions have been supplemented.

References are very numerous and contemporary, which is very good.

Reviewer 2 Report

line 22-38: rewrite the abstract focusing on- background, aims, methods, key results, and conclusion.

Table and Figure numbers should be uniform and according to journal's guidline.

Author Response

Dear Reviewer,

We appreciate your suggestions and insightful comments, as well as the valuable time you have devoted to this work. We have now implemented the suggested modifications, all of which have been included in our revised manuscript submission.

Reviewer #2:

line 22-38: rewrite the abstract focusing on- background, aims, methods, key results, and conclusion.

The abstract has been restructured as requested.

Table and Figure numbers should be uniform and according to journal's guidline.

The required content has been updated in accordance with the journal's guidelines.
